# Unraveling the Gut Microbiota: Implications for Precision Nutrition and Personalized Medicine

**DOI:** 10.3390/nu16223806

**Published:** 2024-11-06

**Authors:** Alessio Abeltino, Duaa Hatem, Cassandra Serantoni, Alessia Riente, Michele Maria De Giulio, Marco De Spirito, Flavio De Maio, Giuseppe Maulucci

**Affiliations:** 1Metabolic Intelligence Lab, Department of Neuroscience, Università Cattolica del Sacro Cuore, Largo Francesco Vito, 1, 00168 Rome, Italy; alessio.abeltino@unicatt.it (A.A.); duaa.hatem@unicatt.it (D.H.); cassandra.serantoni@unicatt.it (C.S.); alessia.riente@unicatt.it (A.R.); michelemaria.degiulio@unicatt.it (M.M.D.G.); marco.despirito@unicatt.it (M.D.S.); 2UOC Physics for Life Sciences, Fondazione Policlinico Universitario “A. Gemelli” IRCCS, 00168 Rome, Italy; 3Department of Laboratory and Infectious Sciences, Fondazione Policlinico Universitario A. Gemelli IRCCS, L.go A. Gemelli 8, 00168 Rome, Italy

**Keywords:** gut microbiota, metabolism, precision nutrition, personalized medicine, microbiota–host interactions

## Abstract

Recent studies have shown a growing interest in the complex relationship between the human gut microbiota, metabolism, and overall health. This review aims to explore the gut microbiota–host association, focusing on its implications for precision nutrition and personalized medicine. The objective is to highlight how gut microbiota modulate metabolic and immune functions, contributing to disease susceptibility and wellbeing. The review synthesizes recent research findings, analyzing key studies on the influence of gut microbiota on lipid and carbohydrate metabolism, intestinal health, neurobehavioral regulation, and endocrine signaling. Data were drawn from both experimental and clinical trials examining microbiota–host interactions relevant to precision nutrition. Our findings highlight the essential role of gut microbiota-derived metabolites in regulating host metabolism, including lipid and glucose pathways. These metabolites have been found to influence immune responses and gut barrier integrity. Additionally, the microbiota impacts broader physiological processes, including neuroendocrine regulation, which could be crucial for dietary interventions. Therefore, understanding the molecular mechanisms of dietary–microbiota–host interactions is pivotal for advancing personalized nutrition strategies. Tailored dietary recommendations based on individual gut microbiota compositions hold promise for improving health outcomes, potentially revolutionizing future healthcare approaches across diverse populations.

## 1. Introduction

In recent years, there has been a growing interest in understanding the complex relationship between the human gut microbiota, metabolism, and overall health. This increased attention highlights the scientific community’s growing recognition of the crucial role that the gut microbiota plays in influencing various aspects of human physiology and health outcomes. This paradigm shift has been fueled by advances in technology, particularly in the field of high-throughput sequencing, which have enabled researchers to delve deeper into the complex ecosystem that exists within the human gut [1]. The human gut, a dynamic and multifaceted ecosystem, serves as a habitat for trillions of microorganisms, collectively known as the gut microbiota [2]. This intricate microbial community, consisting predominantly of bacteria [3], but also encompassing viruses, fungi, and archaea, establishes a symbiotic relationship with its human host, exerting profound effects on various aspects of human health and physiology. The gut microbiota, often aptly described as the “forgotten organ” [4,5], orchestrates a multitude of essential functions vital for maintaining homeostasis and overall wellbeing.

At its core, the gut microbiota is intricately involved in nutrient metabolism, serving as a critical mediator in the digestion, absorption, and utilization of dietary components [6]. Through its diverse enzymatic activities, the microbiota plays a pivotal role in breaking down complex carbohydrates, proteins, and fats that would otherwise be indigestible by the human host. Recent studies have implicated the gut microbiota in the regulation of host metabolism, including energy balance, lipid metabolism, and glucose homeostasis, highlighting its central role in metabolic health [7,8]. Moreover, certain microbial species (e.g., *Escherica coli*, *Bifidobacteria*, and *Lactobacilli*) within the gut microbiota possess the ability to synthesize essential vitamins and other bioactive compounds, further contributing to the host’s nutritional status [9].

Beyond its role in nutrient metabolism, the gut microbiota exerts profound influences on host immune function and homeostasis [10,11]. This microbial community plays a pivotal role in shaping the development and maturation of the host immune system, fostering immune tolerance, and defending against invading pathogens. Additionally, the gut microbiota contributes to the maintenance of intestinal barrier integrity, thereby preventing the translocation of harmful microbes and antigens into systemic circulation [12,13].

Moreover, the symbiotic relationship between the gut microbiota and the host encompasses broader physiological processes, including neurobehavioral regulation and endocrine signaling [14,15]. Mounting evidence suggests that the gut microbiota plays a crucial role in modulating the gut–brain axis, influencing mood, cognition, and behaviour, through intricate signaling pathways [14,15].

Disruptions in the composition and function of the gut microbiota, termed dysbiosis [1], have been linked to a plethora of health conditions [16], spanning metabolic [17], gastrointestinal [18], and even neurological disorders [19]. Dysbiosis is characterized by alterations in the relative abundance and diversity of microbial taxa within the gut ecosystem, often accompanied by functional changes that perturb host–microbe interactions. This association was initially proposed by Metchnikoff in 1907, suggesting that replacing or reducing “putrefactive” bacteria in the gut with lactic acid bacteria could normalize gut health and prolong life [20]. Recent studies strongly suggest that dysbiosis contributes to the development of various disorders, such as irritable bowel syndrome (IBS), intestinal tumors, obesity, and type 1 diabetes (T1D) [21,22,23,24,25,26,27]. It has also been found that *Firmicutes*, one of the most abundant phyla in the human intestine, exhibit reduced community complexity in Crohn’s disease [28]. Furthermore, emerging evidence suggests a potential link between dysbiosis and neurological and psychiatric conditions, including depression, anxiety, and autism spectrum disorders [29,30,31].

Understanding the intricate web of interactions between dietary components, lifestyle factors, and the gut microbiota is not merely an academic pursuit but a critical endeavor with far-reaching implications for advancing precision nutrition approaches [32,33]. Precision nutrition is a tailored dietary strategy that considers an individual’s unique genetic, environmental, and physiological factors to optimize health outcomes and prevent disease [34]. At the heart of precision nutrition lies the recognition that each individual is biochemically unique, shaped by genetic predispositions, environmental exposures, and the dynamic interplay between host physiology and the gut microbiota [35]. Precision nutrition encompasses a holistic approach that extends beyond macronutrient composition to consider micronutrient adequacy, dietary fiber intake, phytochemical diversity, and meal timing—all factors that can profoundly shape the gut microbiota and impact metabolic health [36,37]. By integrating cutting-edge technologies, such as high-throughput sequencing [38], metabolomics [39], and microbiome analysis [40], researchers can delineate the intricate molecular mechanisms underlying dietary–microbiota–host interactions, paving the way for personalized dietary recommendations tailored to an individual’s unique microbial profile [41]. Furthermore, precision nutrition holds immense potential for preventing and managing a myriad of chronic diseases, ranging from metabolic disorders, like obesity and type 2 diabetes (T2D), to gastrointestinal ailments, such as inflammatory bowel disease (IBD) and IBS [42,43]. By leveraging insights into the gut microbiota’s role in disease pathogenesis, clinicians can design personalized dietary interventions that target specific microbial imbalances or dysbiosis patterns, thereby optimizing therapeutic efficacy and minimizing adverse effects [44].

This review provides a thorough examination of how the gut microbiota influences human metabolism, immune function, and disease susceptibility. By analyzing the latest research, we aim to shed light on the implications of this relationship for precision nutrition. Our synthesis of existing evidence aims to advance understanding in this field and pave the way for targeted therapeutic interventions. This review serves as a catalyst for further research and underscores the importance of considering the gut microbiota in healthcare strategies.

## 2. Methods

A gray literature search was carried out to select the most suitable articles for this review. We partly followed the approach used in Soldani et al. [45]. To summarize, we used reference databases, such as PubMed, Scopus, ScienceDirect, and Google Scholar, using the following keywords: “gut microbiota in human health”, “gut microbiota diet”, “microbiome diversity”, and “gut microbiota and precision nutrition”. “Saturation” was used as a stopping criterion: the search was stopped when no new results/relevant concepts emerged from the search results. The inclusion criteria considered were (i) the relevance to the topic, i.e, relevant studies of the microbiome in relation with human health and metabolism in the field of precision nutrition and clinical nutrition and (ii) publication within the last 20 years (2004–2024).

The research was carried out considering both reviews and original articles. The keywords used are related to the microbiome. Different types of studies were used, as in this way, it was possible to obtain a broader overview [45].

## 3. Measurement of Gut Microbiota

Several methods have been developed to analyze the human gut microbiota, each offering unique advantages and limitations. The first important distinction is based on the application of culture-based techniques and/or molecular approaches. Culture and biochemical typing techniques were the gold standard for bacterial species identification for many years [16]. For example, Clostridium difficile can be isolated using specific anaerobic culture conditions with selective media containing fructose and cefoxitin, while Escherichia coli thrives in aerobic conditions with lactose-based media, like MacConkey agar, allowing for differential identification based on substrate fermentation [46,47]. Culture-based techniques have been refined to capture a great number of microorganisms, including anaerobic and aerobic bacteria, that without a host component allows a major reproducibility and a supervised evaluation of more variables. On the other hand, to reach optimal results in terms of taxonomic identification and strain isolation and to facilitate the growth of rare species, several culture conditions are needed: oxygen, pH, temperature, culture medium, and nutrients represent some of the key factors that potentially impact and affect bacterial growth. For this reason, microbial cultures are often lower throughput than molecular methods, as the ultimate culture conditions are not always known for many microorganisms or can require diverse expertise.

Culture-independent techniques have revolutionized our knowledge of the gut microbiota by providing a more representative snapshot of this niche [48,49]. Before next-generation sequencing (NGS) applications, several techniques were commonly used to investigate the composition of a bacterial community, which are further detailed in Table 1 [16].

These techniques rely on small ribosomal RNA subunit (16S rRNA) sequence divergences and can demonstrate the microbial diversity of the gut microbiota, provide qualitative and quantitative information on bacterial species, and detect changes in the gut microbiota in relation to diseases. Examples of these techniques include denaturing gradient gel electrophoresis (DGGE) [50], terminal restriction fragment-length polymorphism (T-RFLP) [51], fluorescence in situ hybridization (FISH) [52], and DNA microarrays [53]. With the emergence and the development of NGS, two applications overcame the others namely 16s rRNA sequencing (targeted-based metagenomics) and whole-genome sequencing (shotgun metagenomics), involving different laboratory organization rather than a diverse bioinformatic complexity [54]. In Figure 1, we schematically report a standard experimental flowchart to obtain the final library pool to sequence.

Basically, targeted-based approaches are defined by the amplification and sequencing of one or more 16s rRNA hypervariable regions, while shotgun sequencing allows for the sequencing of total DNA [55]. On the other hand, shotgun metagenomics is not preceded by a specific amplification, but the total extracted DNA is used to prepare a library to sequence [56,57,58]. The main features of these two applications are schematically summarized in Figure 2.

As various pathologies and diseases are subjected to analysis regarding the potential role of the microbiota in their causation, clinicians and clinical investigators may find themselves bewildered by the proliferation of molecular microbiological approaches to study the microbiota and may struggle to determine the adequacy of a particular methodology to address a specific research question, disease state, or study population. As such, this section presents the techniques currently used to characterize the gut microbiota, suggesting when these techniques may be most appropriately applied in human studies and critically evaluating their advantages and limitations.

## 4. Composition and Function of Gut Microbiota

### 4.1. Dominant Composition and Biogeography of Human Microbiota in the Gastrointestinal Tract

The gut microbiota is characterized by a vast diversity of bacterial species, with four predominant bacterial phyla (Table 2 [59,60,61,62]).

In addition to these predominant phyla, the gut microbiota hosts a wide range of other bacterial species, each with specific functions contributing to the host’s wellbeing. A complete description of these species is beyond the scope of this review but can be retrieved in another study [63].

Recent research indicates that the ratios of these phyla, particularly the Firmicutes/Bacteroidetes (F/B) ratio, serve as important biomarkers of gut dysbiosis and are frequently cited in the literature as indicators of obesity [64,65]. Studies have shown that obese individuals typically exhibit a higher F/B ratio compared to their normal-weight counterparts, suggesting a link between this ratio and metabolic disorders. Conversely, lower F/B ratios have been associated with leaner phenotypes and may reflect a healthier gut microbiome [64].

In clinical settings, the F/B ratio has been correlated with various health outcomes, including inflammation and metabolic syndrome [65]. For instance, increased F/B ratios are observed in patients with cirrhosis, where they correlate with worse prognosis and higher mortality rates [66]. These findings highlight the relevance of the F/B ratio not only as a marker of obesity but also as a potential indicator of overall gut health and disease susceptibility.

In the small intestine, typically high levels of acids, oxygen, and antimicrobials are present, along with a short transit time [67]. These properties limit bacterial growth, so only rapidly proliferating, facultative anaerobic bacteria capable of adhering to the epithelium/mucosa survive [67].

In contrast to varying microbiota compositions among different gastrointestinal organs, the microbiota within specific mucosal regions remains spatially conserved in terms of diversity and composition, even during localized inflammation [68,69]. However, differences emerge between fecal/luminal and mucosal compositions, with *Bacteroidetes* more abundant in fecal/luminal samples and *Firmicutes*, particularly *Clostridium* cluster XIVa, enriched in the mucus layer [70]. Recent mouse experiments show that bacterial species grow and use resources differently in the outer mucus layer versus the intestinal lumen, highlighting the need for careful sampling methods in microbiota analysis [71]. While interindividual differences in species disposition prevail over differences within individuals, the concept of a “core microbiota” remains debated [72,73]. Instead, a functional approach to defining the core microbiota based on microbial gene repertoire is suggested. Microbiotic arrangements have been also divided into predictive “community types”, such as enterotypes, associated with varying backgrounds [74]. However, the existence and formation of these enterotypes remain controversial, subject to ongoing debate in the literature.

### 4.2. Dynamism and Influencing Factors

The human gut microbiota is established in early life but can be subsequently altered by various factors influencing its development and diversity [75]. Understanding these dynamics and influencing factors is crucial for comprehending the role of the gut microbiota in health and disease. The main factors influencing the microbiota are summarized in Table 3 [76,77,78,79,80,81,82,83,84,85,86,87,88,89,90,91,92,93,94,95].

#### 4.2.1. Impact of Dietary Fibers

The intestinal microbiota plays a fundamental role in the fermentation of dietary fibers, contributing to the production of SCFAs and other metabolites that influence host health [96,97]. The large intestine is a continuous and nutrient-rich environment, with various physiological conditions along its different parts. In the ileum, the mucus layer is thin, with rapid transit and low microbial diversity and density, while in the distal tract of the large intestine, the mucus layer is thick, with slow transit and high microbial diversity and density [98]. Dietary fibers, such as cellulose, hemicellulose, and pectin derived from plant cell walls, undergo degradation by the intestinal microbiota, with primary bacteria initiating the degradation process and secondary bacteria utilizing partially hydrolyzed products as substrates [98,99]. Primary bacteria, such as *Eubacterium*, *Roseburia*, *Ruminococcus*, *Clostridium*, and *Bifidobacterium*, are the first to colonize and initiate the degradation of complex fibers. Partially hydrolyzed products from these bacteria are used as substrates by secondary bacteria, contributing to the production of SCFAs and other metabolites. Dietary fibers with greater resistance to fermentation are metabolized more distally in the colon, while those more easily fermentable are used more proximally [100]. The fermentation of fibers by the intestinal microbiota leads to the production of SCFAs, primarily acetate, propionate, and butyrate, which are absorbed by the host and used as an energy source [101] CFAs have several beneficial effects on host health. For example, butyrate is known for its role in preventing colon cancer and strengthening the colonic defensive barrier [102]. SCFAs can also modulate inflammatory processes, influence immune response, and regulate endothelial function [103]. Furthermore, dietary fibers can modulate the composition of the intestinal microbiota, conferring various health benefits to the host. Studies on specific dietary fibers, such as arabinoxylan, xyloglucan, fructo-oligosaccharides, and pectin, have demonstrated their role in modulating microbiota composition and SCFA production, resulting in the improved metabolic and immune health of the host [104,105,106,107]. For example, arabinoxylan promotes the growth of fiber-degrading bacteria, increasing SCFA production and decreasing the abundance of opportunistic pathogens, such as *Desulfovibrio* and *Klebsiella*, thereby improving metabolic health [104]. Xyloglucan supplementation reduces body weight and liver damage in high-fat diet-fed mice by modulating gut microbiota composition and upregulating bile acid metabolism pathways [106]. Similarly, fructo-oligosaccharides (FOS) increase the numbers of *Lactobacillus* and *Bacteroides*, enhancing calcium absorption and reducing systemic inflammation in senescence-accelerated mice [108]. Pectin, particularly citrus pectin oligosaccharides, has hypocholesterolemic effects by increasing beneficial bacterial groups, such as *Bifidobacterium*, *Lactobacillus*, and *Bacteroides*, which correlate with higher SCFA levels and improved cholesterol metabolism [105].

Moreover, dietary fibers contained in whole grains can influence the composition of the intestinal microbiota, fostering the growth of beneficial bacteria. These fibers are complex carbohydrates that undergo fermentation by gut microbes, leading to the production of SCFAs, such as acetate, propionate, and butyrate. These SCFAs play a pivotal role in maintaining intestinal health by reducing inflammation and supporting overall host wellbeing. The fermentation of fibers in whole grains can lead to the production of SCFAs and other metabolites that exert beneficial effects on host metabolic and immune health. Studies in mice and cellular models have shown that supplementation with whole grains can influence the composition of the intestinal microbiota and improve metabolic and immune health [109,110]. For example, a diet based on whole grains is associated with a reduced risk of obesity, heart disease, and T2D, in part due to the positive effects of the intestinal microbiota [111].

#### 4.2.2. Impact of Prebiotics, Probiotics, Postbiotics, and Synbiotics

Prebiotics and probiotics play significant roles in modulating the composition and function of the gut microbiota, thereby influencing host health [112]. Probiotics, defined as live organisms that confer a benefit to the host when provided in adequate quantities, encompass various strains of bacteria, such as *Escherichia coli* Nissle 1917, *Lactobacilli*, and *Bifidobacteria*. These probiotics contribute to gut health by producing SCFAs and inhibiting the growth of pathogenic bacteria through competition for nutrients or receptors on the gut wall [113,114]. Certain bacteria, like *Bifidobacteria* and *Lactobacilli*, are devoid of lipopolysaccharides (LPS), reducing the risk of infection. Additionally, bacterial species, like *Roseburia* [115] and *Akkermansia muciniphila* [116], have been identified as potential probiotics, further expanding the spectrum of beneficial microorganisms. On the other hand, prebiotics are nutrients that promote the growth or activity of specific microbial genera and species in the gut microbiota, conferring health benefits to the host. They stimulate the growth of *Bifidobacteria* and *Lactobacilli*, thereby restoring a healthy gut microbiota composition. Dietary interventions involving prebiotics have shown promise in modulating the gut immune response and restoring intestinal homeostasis [117,118,119]. Recent research has explored novel prebiotics derived from traditional Chinese medicine, such as water mycelium extracts of *Ganoderma lucidum*, *Hirsutella sinensis*, and *Antrodia cinnamomea* [112]. These fungal remedies have demonstrated efficacy in reducing body weight, inhibiting obesity-induced complications, and alleviating inflammation and insulin resistance in high-fat diet mice. Importantly, the effects of these fungal products were mediated through modulation of the gut microbiota, highlighting their potential as prebiotic agents for promoting gut health and overall wellbeing.

Postbiotics are non-viable bacterial products or metabolic byproducts produced by probiotic microorganisms that have biologic activity in the host [120,121]. These functional bioactive compounds are generated during anaerobic fermentation of organic nutrients, like prebiotics, and include short-chain fatty acids, microbial cell fragments, extracellular polysaccharides, cell lysates, teichoic acid, vitamins, and other low molecular weight soluble compounds [122]. Postbiotics have several advantages over live probiotics. They are more stable and safer, as they do not require live microorganisms to confer health benefits. Postbiotics have been shown to enhance gut barrier function, modulate immune responses, and inhibit the growth of pathogenic bacteria [123]. This makes them a promising alternative for managing various health conditions, particularly those related to inflammation and metabolic disorders. Beneficial effects of postbiotics can also be found in food and pharmaceutical applications. Indeed, they can be used as food bio preservatives, in food packaging, and for the biodegradation of food safety-related chemical contaminants. In the pharmaceutical industry, postbiotics have exhibited anti-inflammatory, immunomodulatory, antihypertensive, and antioxidant activities.

Synbiotics are dietary supplements that combine the activity of both probiotics and prebiotics to promote host health synergistically. They can also help strengthen the host immune system and protect against harmful pathogens [124,125].

By nurturing a symbiotic relationship between prebiotics and probiotics, synbiotics hold promise in reshaping the microbial landscape of the gut, thereby offering a multifaceted approach to managing various health conditions. Synbiotics can help reduce inflammation, prevent insulin resistance, and encourage the release of glucagon-like peptide-1 in the host [124].

Prebiotics, probiotics, postbiotics, synbiotics, and recent nutribiotics or pharmabiotics can offer promising strategies for modulating the gut microbiota and promoting host health [126]. Their ability to selectively promote beneficial microbial populations while inhibiting pathogens underscores their potential therapeutic applications in managing and modulating host immune response and metabolic disorders. Further research is warranted to elucidate the mechanisms underlying their effects and to optimize their clinical utility.

#### 4.2.3. Impact of Chewing

Chewing plays a significant role in interacting with the gut microbiota, the collection of bacteria and other microorganisms that populate the gut [127,128]. It is not only integral to the mechanical breakdown of food but is also pivotal in various metabolic processes. Chewing can significantly affect smoking habits by altering the sensory experience and reducing cravings, which is crucial for understanding how behavioral interventions can support smoking cessation programs.

Additionally, the relationship between chewing and glucose metabolism is significant. Chewing stimulates saliva production, which contains enzymes that initiate carbohydrate digestion, thereby influencing glucose levels. This connection has profound implications for managing certain conditions, like diabetes and metabolic syndrome [129].

To deepen our understanding of these interactions, it is proposed to use specialized devices equipped with unique metrics to monitor and evaluate chewing patterns. These devices can provide detailed data on the frequency, duration, and intensity of chewing, which can be correlated with changes in the microbiome, smoking behaviour, and glucose metabolism. Integrating these metrics into research allows for comprehensive studies that assess the direct effects of chewing and explore its broader implications on overall health. This approach can help develop targeted interventions and personalized treatment plans that leverage the benefits of chewing to improve metabolic health and support smoking cessation efforts.

In summary, chewing is an essential part of the digestive process that can significantly influence the composition and function of the gut microbiota. Proper chewing can promote microbial diversity, beneficial metabolite production, and overall intestinal health.

### 4.3. Impact of Gut Microbiota on Other Body Systems

The gut microbiota plays an indispensable role in human health, influencing a wide array of physiological processes and contributing to disease prevention. The multifaceted interactions between the microbiota and the host organism have a significant impact on overall health and can influence the development of pathological conditions. The gut microbiota modulates the host’s immune system by interacting with mucosal and systemic immune cells, thereby influencing immune cell maturation and function [130]. Intestinal bacteria aid in digesting and absorbing nutrients, and they synthesize essential vitamins, such as B12, folic acid, and vitamin K, which are crucial for host metabolic processes [131]. Additionally, the gut microbiota provides a protective barrier against harmful pathogens by competing for resources, producing antimicrobial substances, and modulating host immune responses [132]. Gut microbes also influence nutrient absorption, hormone production related to appetite, and fat deposition, thereby impacting susceptibility to obesity and metabolic diseases [27]. Furthermore, microbiota-derived metabolites, like short-chain fatty acids and polyphenols, have anti-inflammatory and antioxidant effects, protecting against chronic inflammation and associated diseases [133].

Overall, the interactions between the gut microbiota and other body systems, especially the immune, nervous, and endocrine systems, are pivotal for maintaining organismal homeostasis and overall health. Here, a concise overview of these interactions is presented.

#### 4.3.1. Immune System

The immune system, a complex network of cells, tissues, and organs, is the body’s defense mechanism against pathogens and foreign invaders [134]. It consists of two main parts: the innate immune system, which provides immediate defense, and the adaptive immune system, which develops targeted responses. In this section, the intricate interplay between the microbiota—the collection of microorganisms inhabiting the body—and the immune system will be explored. The intricate interplay between the microbiota and the immune system is a dynamic process, heavily influenced by various internal and external factors [135]. Just as the immune system is subject to modulation by age, sex, diet, exercise, and environmental factors, so too is the composition and function of the microbiota within the human body [136]. Understanding this interaction is crucial for elucidating mechanisms of health and disease, as well as for developing targeted therapeutic interventions. The gastrointestinal (GI) tract harbors the highest concentration of microorganisms in the human body, forming a symbiotic relationship with the immune system [137]. This relationship aids in pathogen colonization resistance and influences the body’s response to pathogens and the efficacy of immune responses. The microbiome profoundly affects various aspects of immune function, including cytokine production, maintenance of homeostasis, regulation of T cell production, and modulation of the overall immune system [138,139,140,141]. For instance, microbiota-derived antimicrobial peptides, such as bacteriocins, contribute to pathogen clearance within the microbiome, augmenting the immune response [142]. Environmental factors, particularly during birth and infancy, significantly shape the composition and function of the microbiota, thereby impacting immune development and responsiveness [143]. Antibiotics and dietary patterns also exert notable effects on the microbiome, subsequently influencing immune function [144]. While crucial for combating infections, antibiotics can disrupt the balance and diversity of the microbiota, compromising immune efficacy. Moreover, dysbiosis, characterized by an imbalance in the microbiota, is implicated in various diseases, including IBD, T1D, multiple sclerosis, HIV, and certain cancers [145,146,147]. The microbiota’s involvement in disease pathogenesis is exemplified by its association with conditions, like T1D and colorectal cancer (CRC) [148]. Alterations in gut microbiota composition, particularly during infancy, may predispose individuals to T1D by promoting proinflammatory states. Similarly, the microbiome’s influence on tumor development and progression in CRC underscores its role in cancer pathogenesis. Recognizing the pivotal role of the microbiota in disease pathophysiology has led to novel therapeutic strategies, such as fecal microbiota transplantation (FMT) [149]. FMT, which restores microbial balance in patients with gut dysbiosis, holds promise for treating conditions, like *Clostridioides difficile* infection (CDI), by modulating proinflammatory cytokines and enhancing anti-inflammatory bacteria [150,151]. Emerging research highlights the potential of microbiota-based interventions for personalized medicine. Prebiotics, probiotics, synbiotics, and bacteriophages offer avenues for modulating the microbiome to promote health and mitigate disease risk [152,153]. Moreover, understanding the impact of the microbiome on vaccine efficacy may inform strategies for enhancing immunological memory and improving protection against viral infections [154].

#### 4.3.2. Nervous System

The relationship between the microbiota and the nervous system, known as the microbiota–gut–brain axis, underscores the intricate communication between the gut microbiota and the brain, exerting profound effects on both physical and mental health [155,156]. This bidirectional communication pathway involves various systems, including the central, autonomic, and enteric nervous systems, as well as the immune and endocrine systems. The interaction between the central nervous system (CNS) and the enteric nervous system (ENS) is a complex and bidirectional communication process that significantly influences gastrointestinal functions and mobility. The CNS, comprising the brain and spinal cord, communicates with the gut through efferent and afferent nerves, regulating these functions [157]. The ENS, an intrinsic network of neurons within the gut wall, operates independently but interfaces with the CNS through the vagus nerve and spinal terminals, modulating gut motility, secretion, and sensation [158,159]. The gut microbiota also plays a crucial role by producing neurotransmitters, vitamins, and metabolites, like SCFAs, which can influence neuronal function. Although the blood–brain barrier restricts direct access to the brain, microbial metabolites can activate afferent sensory neurons of the vagus nerve, transmitting signals to the brain via neuroimmune and neuroendocrine pathways [160,161,162]. Germ-free animal studies highlight the importance of gut microbes in brain development and neuroinflammatory responses [163,164]. Dysregulation of this microbiota–gut–brain axis is linked to mental health disorders, such as anxiety, depression, and neurodegenerative diseases [165,166]. Germ-free mice exhibit learning deficits, anxiety-like behaviour, and stress responses, which can be mitigated by microbial colonization [163,164]. Moreover, alterations in gut microbiota composition due to stress, antibiotics, or diet impact brain function and behaviour [162]. Therapeutically, probiotics and prebiotics show potential in modulating gut microbiota and improving brain health, with specific probiotic strains exhibiting antidepressant-like effects and reducing anxiety-related behaviors by influencing neurotransmission and neuroimmune pathways [165,167]. FMT has emerged as a novel therapeutic approach for conditions, such as IBS and Parkinson’s disease, emphasizing the potential of targeting the microbiota–gut–brain axis in disease management [168].

In summary, the microbiota–gut–brain axis represents a complex interplay between the gut microbiota and the nervous system, with profound implications for mental health, behaviour, and disease susceptibility. Further research into the mechanistic underpinnings of this axis holds promise for the development of novel therapeutic strategies targeting the gut microbiota to promote brain health and mitigate neurological disorders.

#### 4.3.3. Endocrine System

Since birth, bacterial colonization of the gut has a role in the maturation of the immune and endocrine systems [169,170]. Remarkably, commensal bacteria have been found to synthesize and release hormones [171], thereby engaging in a dialogue with the host’s metabolism, immunity, and behaviour. This interaction is bidirectional, as host hormones have been demonstrated to both affect and be affected by the microbiota. Lyte and Ernst pioneered the field of endocrine microbiology by demonstrating that stress-induced neuroendocrine hormones can impact bacterial growth [172]. Subsequent investigations in this field have identified hormone receptors in microorganisms, suggesting their involvement in intercellular communication [173]. Other studies have shown that many enzymes involved in host hormone metabolism can be derived from horizontal gene transfers from bacteria [174]. Further clues to the existence of interactions between bacteria and the endocrine system have emerged from the discovery of inter-kingdom signaling, including hormonal communication between microorganisms and their hosts [175]. This field has evolved from the initial observation that bacteria perform quorum sensing (QS), a communication based on the production and detection of autoinducer molecules [176]. These autoinducer molecules are hormone-like elements that regulate functions, including coordinated bacterial growth, motility, and virulence [133]. In addition to influencing bacteria, these signals can modulate host cell signal transduction [177]. Host hormones also influence bacterial gene expression, which in turn can have consequences for their hosts. For example, catecholamines enhance bacterial adhesion to host tissues and influence bacterial growth and virulence [178,179]. Conversely, human sex hormones estrone and estradiol reduce bacterial virulence by inhibiting QS [180]. The effects of host hormones on the microbiota are manifold and include bacterial growth, virulence, and resistance. Variations in hormone levels resulting from host factors, such as diet, exercise, mood, overall health status, stress, and sex can influence intestinal microbial balances. These hormone–microbiota interactions influence a wide range of host responses, including behaviour, metabolism, appetite, and immune responses.

In a recent study, it was examined whether the membrane fluidity of red blood cells (RBCs), influenced by various pathways induced by hyperglycemia, could provide a complementary index of HbA1c to monitor the development of macroangiopathic complications related to T2D, such as peripheral arterial disease (PAD) [181]. Altered RBC membrane fluidity is associated with a spatial reconfiguration of liquid crystal (LC) domains, associated with the development of T2D-related macroangiopathic complications. These findings are consistent with recent discoveries in the field of endocrine microbiology, highlighting a complex interaction between the gut microbiota and the endocrine system, with significant implications for health and disease [171].

While this field is still in its early stages, future research will likely identify further significant interconnections between hormones and the microbiome. Endocrine microbiology may also explain how the microbiota influences the gastrointestinal and psychological health of the host. As such, hormones represent an important mechanism for host–microbiota interaction.

## 5. Human Metabolism and Gut Microbiota

The gut microbiota actively participates in the metabolism of a wide range of nutrients obtained from the daily diet, including lipids, proteins, and carbohydrates [89]. Upon metabolism by the gut microbiota, these nutrients generate a series of bioactive metabolites that directly influence the host’s metabolism and health [182].

The complex interaction between gut microbiota and human health involves various bacterial species, their metabolites, and their effects on host physiology. The gut microbiota comprises diverse bacterial species, including Akkermansia (Gram-negative, anaerobic), Roseburia (Gram-positive, anaerobic), non-pathogenic Escherichia coli, Bifidobacteria (Gram-positive, obligate anaerobes), and Lactobacilli (Gram-positive, facultative anaerobes). These bacteria produce several important metabolic compounds, including the following:Short-chain fatty acids (SCFAs), such as acetate, propionate, and butyrate.Amino acid metabolites, like ammonia and indole.Essential vitamins, particularly vitamin K- and B-group vitamins.Postbiotics, including extracellular polysaccharides.Fermentation metabolites, such as lactate and glycerol.Various bioactive components, including bacteriocins.

These bacterial metabolites significantly influence human health through multiple pathways, particularly in terms of regulating metabolism (lipid, carbohydrate, and protein metabolism), maintaining intestinal barrier function, modulating inflammation, and regulating immune responses (Figure 3).

This section provides a detailed exploration of key metabolites produced by the gut microbiota, highlighting their interactions with human metabolic processes and their roles in health promotion and disease prevention.

These compounds, primarily stemming from the fermentation of indigestible substrates, encompass SCFAs [183], metabolites derived from amino acid metabolism [184], and essential vitamins [185]. Specifically, SCFAs emerge as central players, originating from the fermentation of non-digestible carbohydrates. Acetate, propionate, and butyrate, the primary SCFAs produced, not only serve as an energy source for intestinal epithelial cells, but also modulate inflammation and regulate host lipid and carbohydrate metabolism [186,187]. Additionally, the gut microbiota is responsible for the production of lipopolysaccharides (LPS) and bile acid conjugates, which can influence intestinal inflammation and lipid absorption [188,189]. Concurrently, metabolites derived from amino acid metabolism, such as ammonia, indole, and sulfides, exhibit a dual nature, potentially protective or detrimental to human health [190]. While indole and its derivatives are associated with immune system regulation and cancer prevention [191,192], elevated levels of ammonia and sulfides can prove toxic to intestinal epithelial cells [193,194,195]. Furthermore, the production of vitamins, including vitamin K and select B vitamins, plays an essential role in blood clotting and the human energy metabolism [196,197].

The dynamic interaction between microbiota metabolites and human metabolic processes (Figure 3) occurs through complex mechanisms. For instance, SCFAs influence gene expression in the human host, modulating the enzymatic activity involved in lipid and carbohydrate metabolism [198]. Additionally, these metabolites can modulate the composition and function of the microbiota itself, creating an intricate metabolic cycle between the microbiota and the human host (Figure 3).

The following sections will provide a detailed description of the primary metabolites produced by the gut microbiota, along with their direct effects on host metabolism and health. These metabolites will be differentiated according to their roles in the regulation of lipid, protein, and carbohydrate metabolism.

### 5.1. Regulation of Lipid Metabolism

Early studies comparing germ-free mice with conventionally raised mice provided initial evidence for the influence of gut microbes on host energy metabolism and lipid levels [199,200]. While these studies highlighted the impact of gut microbiota on lipid metabolism, they lacked the ability to pinpoint specific microbial candidates responsible for the observed phenotypic changes in conventionally colonized mice. In humans, clinical correlations between obesity, metabolic disorders, and dyslipidemia suggest potential associations between gut bacterial taxa and lipid levels. Recent analyses in population-based cohorts have not only reaffirmed known associations between obesity and specific bacterial taxa (e.g., *Akkermansia* (N34), *Christensenellaceae* (phylum *Firmicutes*; N18), *Tenericutes* (order RF-39; N33), *Eggerthella* (N3), and *Butyricimonas* (N9)), but have also unveiled associations between microbial composition and lipid levels independent of body mass index (BMI) [201].These bacterially derived bile acids can influence hepatic and systemic lipid and glucose metabolism through receptors, such as the farnesoid X receptor (FXR) or G-protein-coupled bile acid receptor 1 (TGR5), thereby impacting host lipid levels [202,203,204]. Anaerobic bacteria in the cecum and proximal colon ferment nondigestible carbohydrates, yielding SCFAs as metabolites. SCFAs play roles in regulating intestinal immune homeostasis, serving as energy sources for colonic epithelial cells, and inducing intestinal gluconeogenesis [205]. Moreover, SCFAs exert metabolic benefits by affecting energy expenditure and insulin sensitivity through G protein-coupled receptors (GPCRs) [206,207]. Gut bacteria may produce intermediate precursors metabolized by the host into products directly influencing lipid levels. For instance, gut microbe-mediated metabolism of dietary choline and L-carnitine leads to the production of trimethylamine (TMA), subsequently oxidized to trimethylamine N-oxide (TMAO) [208,209,210]. Elevated TMAO levels have been linked to atherosclerosis, suggesting a potential role in lipid metabolism regulation. These mechanisms underscore the intricate interplay between the gut microbiota and lipid metabolism, providing insights into potential therapeutic targets for metabolic disorders.

### 5.2. Regulation of Protein Metabolism

Recent metagenomic studies have illuminated the metabolic capacity of the human gut microbiota, particularly in relation to nitrogenous components and amino acid-related compounds [211,212]. These studies have uncovered a significant enrichment of genes in the human gut microbiome associated with amino acid metabolism compared to the human genome. Consequently, the gut microbiota extends the host’s metabolic capabilities, leading to the synthesis of a diverse array of metabolites, including essential amino acids that humans cannot biosynthesize [213]. The metabolic phenotype (e.g., obese vs. lean) and dietary factors have been found to influence the composition and functional capacity of the human gut microbiota, particularly its amino acid metabolism [214]. Both animal- and plant-based diets exert distinct effects on the expression of genes involved in amino acid metabolism. Animal-based diets tend to upregulate catabolic amino acid genes, whereas plant-based diets increase the expression of biosynthetic pathways for these amino acids [214]. Metabolomic studies have further elucidated the role of the human gut microbiota in amino acid metabolism and its implications for health. Branched-chain amino acids (BCAA) and aromatic amino acids have been linked to obesity and insulin resistance, with microbiota-derived metabolites potentially contributing to these conditions [215]. High-protein diets increase the availability of undigested protein in the large intestine, leading to alterations in microbially derived metabolites, such as branched-chain fatty acids and ammonia, with potential implications for metabolic health [212]. Human studies have demonstrated modifications in microbiota composition and metabolite profiles in response to high-protein diets [216,217,218,219,220,221].

### 5.3. Regulation of Carbohydrate Metabolism

The human gut microbiota plays a crucial role in carbohydrate metabolism, particularly in the breakdown of complex carbohydrates [222,223]. Some studies [222,224] have provided significant insights into this area. Anaerobic microorganisms in the gastrointestinal tract coordinate the breakdown of plant cell walls, utilizing catalytic domains and substrate-binding modules to degrade plant polysaccharides [225]. These anaerobic microorganisms include bacteria, such as *Ruminococcus flavefaciens*, *Prevotella bryantii*, *Fibrobacter succinogenes*, and *Ruminococcus albus*, as well as fungi and protozoa, such as *Polyplastron multivesiculatum*. The specificity and organization of these enzymes vary among different species, defining their ecological niche within the gut community based on the substrates ingested by the host. Studies have shown that certain uncultured bacterial species are closely associated with fibrous substrates in the gut, with specific groups, like *Ruminococcus*-related species, being primary colonizers of insoluble substrates [223,225]. The breakdown of carbohydrates by gut microbiota members (e.g., *Bacteroides*, *Prevotella*, *Veillonella*, *Bifidobacterium*, *Lactobacillus*, *Enterococcus*, *Methanobrevibacter*, and the *Eubacterium cylindroides* group) not only influences energy extraction but also contributes to the production of SCFAs, crucial for colonic health [226]. Butyrate-producing bacteria (e.g., genus *Clostridium*) are integral to colonic health in humans, with diverse phylogenetic groups contributing (such as *Cloistridiales*) to butyrate synthesis [227,228]. These bacteria utilize different gene organizations in the central pathway of butyrate synthesis, generating energy through substrate-level phosphorylation and proton gradients. The microbiota’s ability to process complex carbohydrates has been studied extensively, comparing gut isolates from various environments to understand their survival mechanisms. Studies in animal models have demonstrated the adaptation of gut bacteria to each other, highlighting their specialization and interdependence in carbohydrate metabolism [223]. The functional biodiversity of the gut microbiota in carbohydrate breakdown has only recently been explored. Carbohydrate-degrading populations, particularly butyrate producers, are predominant in healthy individuals, contributing to normal gut fermentation, energy extraction, and host health [108]. However, significant interindividual differences exist in these functional groups.

In conclusion, the human gut microbiota is intricately involved in carbohydrate metabolism, impacting energy extraction, SCFAs production, and overall gut health. Advances in technology have opened new avenues to unravel the complex interactions between dietary carbohydrates, gut microbiota composition, and host metabolism, offering promising prospects for targeted dietary interventions and therapeutic strategies.

### 5.4. Summary

To provide a comprehensive overview of these findings, Table 4 summarizes key insights from recent research on the role of the gut microbiota in metabolic processes.

Each section highlights the mechanisms underlying microbial metabolism, clinical implications, recent advancements, and potential therapeutic applications. This summary aims to offer a consolidated understanding of the multifaceted relationship between the gut microbiota and host metabolism, underscoring the significance of microbial contributions to human health and disease.

## 6. Precision Nutrition and Microbiota Interventions

Increasing attention has been focused on the importance of modulating the gut microbiota through dietary choices and nutritional interventions [32,41,229,230,231,232]. This approach, known as gut microbiota modulation, aims to promote a balanced bacterial composition that supports host health. One of the most effective strategies for modulating the gut microbiota is adopting a fiber-rich diet. Dietary fibers serve as the primary substrate for beneficial gut bacteria, promoting their growth and activity. Sources of fiber, such as fruits, vegetables, legumes, and whole grains, provide a favorable environment for SCFA-producing bacteria, which are essential for intestinal health [233]. Additionally, incorporating probiotics and fermented foods into the diet can aid in restoring the balance of the gut microbiota [234]. Probiotics are strains of beneficial bacteria that, when consumed in adequate amounts, can improve microbiota composition and promote intestinal health. Certain foods, such as yogurt, kefir, sauerkraut, and kimchi, are rich in natural probiotics and can be incorporated into the diet to support a healthy gut microbiome.

However, it is also important to limit the consumption of foods high in added sugars and saturated fats, which may promote the growth of pathogenic bacteria in the gut microbiota [32]. Excessive consumption of these foods can contribute to intestinal inflammation and increase the risk of metabolic disorders.

Integrating prebiotic-rich foods into the diet, such as garlic, onions, Jerusalem artichokes, and other non-digestible fiber-rich vegetables, can selectively promote the growth of beneficial bacteria in the colon, thus helping to maintain a balanced gut microbiota [235]. The Mediterranean diet, characterized by a wide variety of nutrient-rich foods, such as fruits, vegetables, fish, whole grains, and vegetable oils, has been associated with a healthy composition of the gut microbiota and numerous health benefits [236]. This dietary pattern provides a diverse range of nutrients and bioactive compounds that can positively influence intestinal health and microbiota diversity (Table 5 [32,41,236,237,238,239,240,241,242]).

### 6.1. Scientific Evidence on the Efficacy of Interventions

The critical analysis of the available scientific evidence regarding the efficacy of various dietary and nutritional interventions in modulating the gut microbiota plays a crucial role in guiding clinical practices and directing personalized dietary recommendations. Numerous clinical and experimental studies have investigated the impact of diet on the composition and functioning of the gut microbiota [41,243,244,245]. Among the relevant research, one study delved into the complex mechanisms regulating the effects of nutrients and specific foods on the balance and functioning of individual gut microbiota. Through the analysis of seven volunteers, Bianchetti et al. provided valuable insights into the effectiveness of specific personalized dietary interventions in modulating the gut microbiota, contributing to the field of personalized nutrition [41]. Other studies have examined the role of diet in modulating the gut microbiota concerning health and disease [246,247,248]. Evidence suggests that diet can influence the diversity, population size, and metabolic functions of the gut microbiota [249]. However, conducting further longitudinal studies is fundamental to fully understanding the long-term effects of specific diets and dietary components on the gut microbiota [41], as well as to identify variations among individuals [41].

Some studies have investigated how probiotics, prebiotics, and dietary fibers influence the gut microbiota and human health outcomes. For instance, in a recent study [250], Bumjo Oh et al. found that probiotic supplementation significantly increased the abundance of beneficial bacteria, such as *Lactobacillus*, *Bifidobacterium*, and *Bacteroides*. These changes are crucial, as they contribute to a healthier microbiome balance, potentially enhancing immune function and metabolic health. Similarly, another study [251] explored the effects of probiotic yogurt on patients with IBD. Results showed an increase in *Lactobacillus* and *Bifidobacterium* levels in the intestines and colons of IBD patients, suggesting a beneficial impact on gut health. In the context of infant health, research [252] on preterm infants indicated that prebiotic and probiotic supplementation reduced excessive crying and fussing. This improvement was associated with lower levels of potentially harmful bacteria, like *Clostridium histolyticum* group, indicating a positive modulation of the microbiome towards a less colic-prone state. Furthermore, dietary fiber interventions were examined in healthy adults in a study [253]. Although overall microbial community structure did not change significantly, there was a notable increase in beneficial bacteria, such as *Alloprevotella*, *Parabacteroides*, and *Parasutterella*. This shift suggested a promotion of a more balanced microbiota composition, potentially contributing to better overall gut health.

These findings underscore the potential of probiotics, prebiotics, and dietary fibers in optimizing gut microbiota composition for improved health outcomes. Understanding individual variations in gut microbiota response is crucial for tailoring personalized nutrition strategies that leverage the microbiome as a biomarker, as highlighted in recent research [89,254]. This approach opens avenues for developing more effective dietary interventions and preventive measures tailored to individual microbiome profiles [255].

### 6.2. Challenges Associated with Implementation

Implementing dietary and nutritional intervention strategies to modulate the gut microbiota faces various challenges and limitations that influence their effectiveness and practical applicability. One of the main challenges is the individual variability in the effectiveness of intervention strategies. As highlighted by various sources, including clinical studies and longitudinal analyses, gut microbiota responses to dietary interventions can vary significantly from person to person. This variability may be influenced by genetics, environmental factors, and even by the frequency and intensity of the interventions themselves [254,256]. The intrinsic complexity of the gut microbiota represents another fundamental challenge. The microbiota is a complex and dynamic ecosystem characterized by a vast diversity of bacterial species and intricate interactions among them and with the human host. A comprehensive understanding of this system requires multidisciplinary approaches and advanced methodologies, which often prove burdensome and complex to implement. Furthermore, assessing the long-term effects of dietary interventions on the gut microbiota presents significant challenges. Long-term studies are essential to fully understand the lasting impacts of dietary modifications on the microbiota and host health. However, such studies require considerable resources and time and may be subject to follow-up losses and variations in participant compliance over time. Finally, interpreting microbiotic data and translating the information obtained into personalized dietary recommendations remains an important challenge [257]. Although technology has enabled significant advancements in characterizing the gut microbiota, the correlation between microbiota composition and host health is not always direct or linear. It is necessary to develop more sophisticated analytical and interpretative approaches to translate microbiotic data into clinically meaningful and applicable information in clinical practice. A summary of these challenges is reported in Table 6.

In conclusion, the challenges associated with implementing dietary and nutritional intervention strategies for the gut microbiota require a holistic and interdisciplinary approach. It is crucial to address these challenges through collaboration among scientists, clinicians, nutritionists, and researchers to develop more effective and personalized approaches for modulating the gut microbiota and improving host health.

## 7. Conclusions

The advent of advanced technologies, particularly high-throughput sequencing, has facilitated deeper exploration of the intricate relationship between the human gut microbiota, metabolism, and overall health, shedding light on its profound implications for precision nutrition [232,258]. Therefore, understanding the composition and functionality of the gut microbiota, as outlined in Section 4 and Section 5, provides critical insights into its role in human health and disease.

The influence of key metabolites produced by the gut microbiota, such as SCFAs, vitamins, and other bioactive compounds, extends far beyond simple digestive processes. These metabolites are not just byproducts of fermentation; they are essential mediators that profoundly impact host physiology and wellbeing. Moreover, the gut microbiota and its metabolites impact neurobehavioral regulation and endocrine signaling, influencing mood, cognition, and behavior via the microbiota–gut–brain axis and hormonal interactions. Furthermore, precision nutrition uses microbiota insights to tailor diets that support gut health, emphasizing fiber-rich foods, probiotics, and fermented items to boost beneficial bacteria and SCFA production. Reducing sugars and saturated fats lowers pathogenic bacteria, while certain foods, like garlic and artichokes, selectively promote healthy bacteria. Certain diets, like the Mediterranean diet, are rich in diverse, whole foods and enhance microbiota diversity and overall health, though individual variability and the need for long-term studies remain key challenges.

Recent advances using stable isotope probing and metabolite analysis have elucidated microbial metabolites derived from carbohydrates in the human gut. This method offers insights into label incorporation into microbial biomass and metabolites, enabling the study of temporal modulation of the human gut microbiota by individual nutrients. Such studies manipulate substrate availability and microbiota composition to control profiles of short-chain fatty acids. Additionally, this approach provides a quantitative understanding of microbial metabolites’ caloric contribution to host energy uptake [259].

These insights are pivotal for developing targeted therapeutic strategies for metabolic and inflammatory conditions, advancing precision medicine in metabolic health. However, integrating this knowledge into practice through digital platforms underscores the necessity for longitudinal studies to comprehensively assess sustained effects and individual responses to dietary interventions.

The microbiota can be of use as part of the practical solutions utilized by specialists, leveraging platforms, like web-based nutrition management systems, where nutritionists utilize microbiota data to create tailored diets [260]. This integration is further enhanced by the advent of new smart technologies that allow the seamless integration of multiple devices, streamlining data collection and analysis processes. Such advancements enable a more efficient and effective utilization of microbiota insights in practical settings, ultimately contributing to improved health outcomes and personalized dietary interventions [261]. Personalized dietary recommendations can be tailored to optimize therapeutic efficacy and promote overall wellbeing. Proper utilization of microbiota insights holds immense potential for revolutionizing healthcare strategies and improving health outcomes in diverse populations. An effective avenue involves integrating and managing the microbiome within novel metabolic models [262,263,264], which serve as essential tools for crafting personalized dietary interventions. These models must prioritize precision and individualization, rendering them valuable assets in both clinical and nutritional domains. In contrast to overly intricate models, such as genome-scale metabolic models [265,266], which may lack practical applicability due to their complexity, emerging metabolic models should leverage microbiota data to furnish tailored dietary recommendations that correspond to individual metabolic profiles. Moreover, recent studies have highlighted how insights from the gut microbiota can enhance athletic performance by refining VO2max models for accurate assessment of aerobic capacity [267]. Integrating microbiota data enables the development of personalized predictive models and tools [268,269], optimizing training regimens and mitigating injury risks. Overall, this integration revolutionizes nutrition and athletic preparation, fostering a deeper understanding of health and performance optimization.

## Figures and Tables

**Figure 1 nutrients-16-03806-f001:**
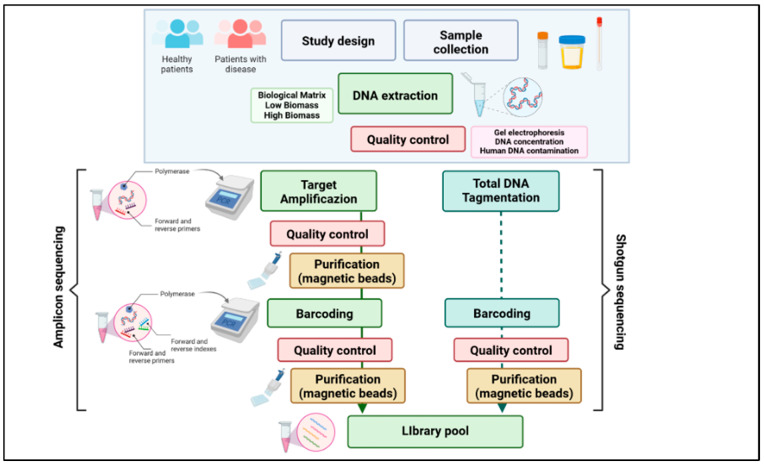
Standard experimental flowchart to obtain the final library pool to sequence. The diagram illustrates the steps involved in both amplicon sequencing and shotgun sequencing. Starting from study design and sample collection from healthy patients and patients with disease, the process includes DNA extraction and quality control. Amplicon sequencing involves target amplification, purification, barcoding, and library pooling, while shotgun sequencing involves total DNA tagmentation, barcoding, and library pooling. Quality control is a critical step throughout both processes.

**Figure 2 nutrients-16-03806-f002:**
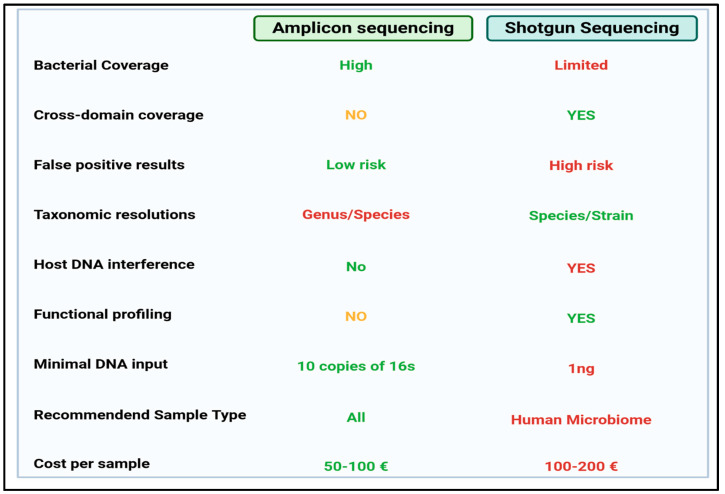
Main features of 16s rRNA amplicon-based and shotgun sequencing approaches. Green and red colors indicate the characteristic that one of the two methods shows as an advantage. In yellow are the features that can be potentially investigated: targeted amplification is able to study other domains (for example mycobiota) if another gene marker is used (ITS region), or a predictive functional profile may be reached.

**Figure 3 nutrients-16-03806-f003:**
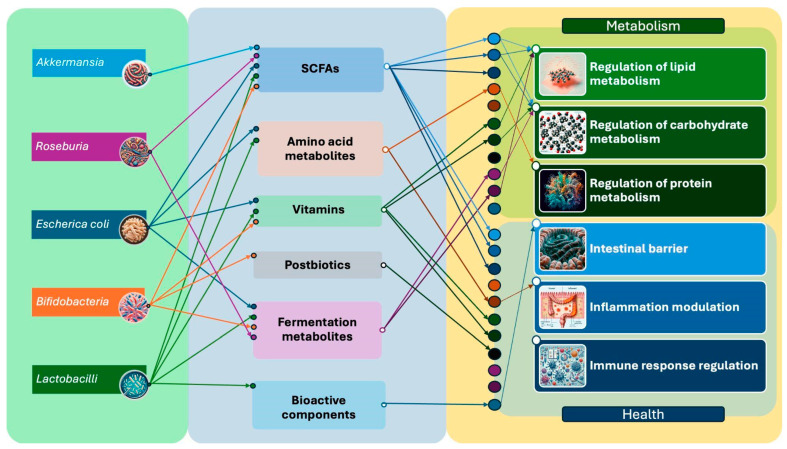
Schematic representation of the interactions between gut microbiota, their metabolites, and their effects on human metabolism and health.

**Table 1 nutrients-16-03806-t001:** Techniques for gut microbiota characterization [16].

Technique	Description	Applications
Culture techniques	Historical approach isolating and growing bacteria in selective culture media. Offers a limited view of diversity and is labor-intensive.	Historical analysis, basic research
16S rRNA	Utilizes conserved the 16S rRNA gene for phylogenetic identification; commonly used in non-culture techniques.	Microbial diversity, taxonomic studies
PCR	Amplifies DNA segments for analysis but is susceptible to bias; used alone or with qPCR for quantification.	Genetic analysis, quantification studies
DGGE and TGGE	DNA fingerprinting techniques separating DNA fragments based on sequence variation; semi-quantitative.	Comparative studies, community profiling
T-RFLP	Fragmentation of 16S rRNA gene amplicons for microbial diversity assessment; semi-quantitative.	Diversity analysis, community comparisons
FISH	Directly visualizes bacterial cells using fluorescent probes; semi-quantitative.	Microbial abundance, spatial distribution
DNA microarrays	Utilizes DNA chip technology for phylogenetic identification; semi-quantitative.	Comparative studies, large-scale analysis
Sequencing	Gold standard for taxonomic identification through 16S rRNA gene sequencing; expensive but comprehensive.	Taxonomic profiling, metagenomic analysis
Massively parallel sequencing	High-throughput sequencing for detecting low-abundance bacteria; provides quantitative data.	Diversity assessment, metagenomic studies
Shotgun sequencing and metagenomics	Analyses genetic and functional diversity of microbiota through random DNA sequencing; comprehensive but costly.	Functional profiling, disease association

**Table 2 nutrients-16-03806-t002:** Predominant bacterial phyla in the gut microbiota: key descriptions and functions.

Phylum	Description	Key Functions	References
* Firmicutes *	One of the most abundant phyla in the human intestine, crucial for gut health.	- Carbohydrate digestion - Production of short-chain fatty acids (SCFAs)	[59]
* Bacteroidetes *	Important for immune system modulation and metabolic processes.	- Degradation of complex polysaccharides - Production of molecules influencing immune system and metabolism	[60]
* Actinobacteria *	Includes *Bifidobacteria*.	- Fermentation of complex carbohydrates - Production of organic acids beneficial for host health	[61]
* Proteobacteria *	Contains both pathogenic and commensal species.	- Vitamin synthesis - Nutrient absorption	[62]

**Table 3 nutrients-16-03806-t003:** Main factors influencing the dynamics and composition of the gut microbiota.

Factor	Description	References
Ethnicity	Specific members of the gut microbiota as mediators of health and diseases can vary based on ethnicity	[76,77]
Genotype	Sex- and age-linked differences have been observed in the gut microbial composition that appear independent of different ethnicities.	[78,79]
Gender	Compared to men, the gut microbiota of premenopausal women exhibited higher microbial diversity and higher abundances of multiple species known to have beneficial effects on host metabolism.	[80]
Age	Age-related gut microbial characteristics have been detected in taxa members of the oral bacterial community.	[80]
Pregnancy	Changes in vaginal microbiota before pregnancy can impair fertility. The composition of the maternal gut microbiome contributes to obstetric outcomes with long-term health sequelae for both mother and child.	[81,82,83]
Mode of delivery	Vaginally born infants have higher levels of Bacteroides, Prevotella, and Lactobacillus in their gut microbiota compared to cesarean-born infants. Cesarean-born infants may still acquire Bacteroides from the maternal gut, not the vaginal microbiome.	[84]
Feeding	Infants’ intestinal colonization may begin before birth, influenced by microbiota from amniotic fluid and placenta. Breastfed infants have a microbiota dominated by Lactobacillus and Bifidobacterium, supported by breast milk oligosaccharides. Formula-fed infants have a microbiota with different dominant species, lacking the same bacterial composition as breastfed infants.	[85,86]
Diet	Diet can induce temporary shifts in gut microbiota; however, it is uncertain if prolonged dietary changes can lead to permanent alterations.	[87,88,89]
Medications and antibiotics	Antibiotics and certain common drugs (e.g., proton-pump inhibitors, metformin, and laxatives) disrupt gut microbiota by reducing species diversity, altering metabolism, and promoting antibiotic-resistant organisms, leading to certain issues, like antibiotic-associated diarrhea and Clostridioides difficile infections.	[90,91,92]
Environmental and lifestyle	Both macroenvironmental (e.g., toxic exposure, socioeconomic status) and microenvironmental factors (e.g., smoking, diet, stress) can disrupt microbiome composition, promoting inflammation and long-term disease development.	[93,94,95]

**Table 4 nutrients-16-03806-t004:** This table provides a concise summary of the key aspects discussed in each section, highlighting the role of gut microbiota, underlying mechanisms, clinical implications, research advances, and potential therapeutic applications.

Aspect	Lipid Metabolism	Protein Metabolism	Carbohydrate Metabolism	Impact of Dietary Fibers	Prebiotics, Probiotics
Role of the gut microbiota	Influences host energy metabolism and lipid levels	Extends host metabolic capabilities, synthesizes diverse array of metabolites	Plays a crucial role in carbohydrate breakdown, energy extraction, and SCFA production	Contributes to the production of SCFAs and other metabolites, modulates microbiota composition	Modulates gut microbiota composition, inhibits pathogenic bacteria growth, restores gut homeostasis
Mechanisms	Production of SCFAs, modulation of bile acid metabolism, influence on lipid levels	Enrichment of genes associated with amino acid metabolism, synthesis of essential amino acids, metabolite production	Coordination of plant cell wall breakdown, production of SCFAs, energy harvesting from different polymers	Degradation of dietary fibers, fermentation leading to SCFA production, modulation of microbiota composition	Production of vitamins, antioxidants, and SCFAs; inhibition of pathogenic bacteria growth, restoration of gut health
Clinical implications	Correlations with obesity, metabolic disorders, dyslipidemia; potential therapeutic targets	Links to obesity, insulin resistance; implications for metabolic health	Links to obesity, insulin resistance; implications for metabolic health	Reduced risk of obesity, heart disease, T2D; improved metabolic and immune health	Therapeutic applications in managing inflammatory diseases, metabolic disorders
Research advances	Population-based cohort analyses, identification of microbial candidates responsible for lipid metabolism	Identification of metabolic capacity of gut microbiota concerning nitrogenous components	Whole-genome sequencing, understanding of polysaccharide utilization mechanisms, metabolite analysis	Studies on specific dietary fibers and their role in modulating microbiota composition and SCFA production	Exploration of novel prebiotics from traditional medicine, effects on gut microbiota and host health

**Table 5 nutrients-16-03806-t005:** Overview of dietary strategies for modulating gut microbiota composition and health.

Dietary Strategy	Characteristics	Benefits	References
Fiber-rich diet	- Rich in dietary fibers sourced from fruits, vegetables, legumes, and whole grains. - Dietary fibers serve as the primary substrate for beneficial gut bacteria, promoting their growth and activity. - Sources of fiber provide a favorable environment for SCFA-producing bacteria essential for intestinal health.	- Promotes growth and activity of beneficial gut bacteria.	[237]
Probiotics and fermented foods	- Incorporates probiotic-rich foods, such as yogurt, kefir, sauerkraut, and kimchi, into the diet. - Fermented foods aid in restoring the balance of the gut microbiota. - Probiotics are beneficial bacteria that improve microbiota composition and promote intestinal health when consumed adequately.	- Aids in restoring balance to the gut microbiota. - Improves microbiota composition and promotes intestinal health.	[238,239]
Limiting sugars and saturated fats	- Reduces consumption of foods high in added sugars and saturated fats. - Limits the growth of pathogenic bacteria in the gut microbiota. - Reducing intake of these foods helps prevent intestinal inflammation and decreases the risk of metabolic disorders.	- Reduces the growth of pathogenic bacteria in the gut microbiota. - Prevents intestinal inflammation and lowers the risk of metabolic disorders.	[240,241]
Incorporating prebiotic-rich foods	- Integrates prebiotic-rich foods, such as garlic, onions, Jerusalem artichokes, and non-digestible fiber-rich vegetables, into the diet. - Prebiotics selectively promote the growth of beneficial bacteria in the colon. - Non-digestible fibers provide a substrate for beneficial bacteria, helping maintain a balanced gut microbiota.	- Selectively promotes growth of beneficial bacteria in the colon. - Helps maintain a balanced gut microbiota.	[242]
Mediterranean diet	- Characterized by an abundance of nutrient-rich foods, such as fruits, vegetables, fish, whole grains, and vegetable oils. - Moderate consumption of dairy products and poultry. - Limited intake of red meat and processed foods. - Provides diverse nutrients and bioactive compounds that positively influence intestinal health and microbiota diversity.	- Associated with a healthy composition of the gut microbiota. - Linked to numerous health benefits including reduced risk of cardiovascular diseases, improved cognitive function, and longevity. - Provides diverse nutrients and bioactive compounds that positively influence intestinal health and microbiota diversity.	[32,41,236]

**Table 6 nutrients-16-03806-t006:** Summary of the challenges associated with implementing dietary and nutritional intervention strategies for modulating the gut microbiota.

Challenges	Description
Individual variability	Gut microbiota responses to dietary interventions vary significantly among individuals due to genetic, environmental, and intervention-related factors.
Complexity of the gut microbiota	The gut microbiota is a complex and dynamic ecosystem with diverse bacterial species and intricate interactions, requiring multidisciplinary approaches for comprehensive understanding.
Long-term effects assessment	Assessing the long-term impacts of dietary interventions on the gut microbiota and host health requires extensive resources, time, and may face challenges, like participant compliance.
Data interpretation and translation	Despite technological advancements, translating microbiota data into personalized dietary recommendations is challenging due to the complexity of the microbiota–host health correlation.
Standardization of methods and practices	Establishing standardized methods (sequencing, analysis, and the type of data shared with clinicians/patients) is essential for creating a reliable and consistent signature of a healthy gut microbiota. This standardization will enhance comparability across studies and improve the reproducibility of results.

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
