# Peer review of "Unraveling the Gut Microbiota: Implications for Precision Nutrition and Personalized Medicine"

_nutrients, 2024, doi:10.3390/nu16223806_

Round 1
Reviewer 1 Report
Comments and Suggestions for Authors
Minor suggestions:
1. Abstract - I suggest the authors change the structure of the abstract into a coherent text without introducing additional sections.
2. line 67 – add citation
3. line 85 – citation after the end of the sentence
4. line 91 – add space before the citation
5. line 97 – authors should use full name type 2 diabetes Mellitus if the letter M appears in the abbreviation
6. line 190 - I suggest removing the quote here; it is still repeated in Table 3
7. line 223 – as above
8. line 634 – as above
9. table 5 - I suggest adding a separate column for citations
10. line 668 – change flora into the microbiome
11. line 680 - note as in the case of other tables regarding citations
12. line 700 – move the citation to the end of the sentence
13. section 6.2 – lack of proper title for the section
Major suggestions:
1. line 124-136 – I would suggest that the authors enrich the description with specific examples, allowing for the identification of microorganisms, taking into account the type of substrate and its differentiating changes
2. table 4 - the table should be a coherent summary of the content, allowing for efficient reading of the content; here in the table, we have collected text, which could be subsequent paragraphs. I suggest the authors change the table's structure and shorten the text or remove it and present the content in the form of text.
3. Line 233 – 236 - the text seems to be written using AI, I suggest the authors to change the language to a less writerly and simpler one
4. Section 4.4 - I suggest moving the section to the beginning of section 4.3 as an introduction, it is more of an introduction or summary of section 4.3 than a standalone chapter 4.4
5. Figure 2 - I appreciate the desire to present such a large amount of content in one graphic, but it is illegible, especially concerning metabolism. In addition, the name of the graphic itself is extended to the level of a separate paragraph and should be shortened. I suggest the authors simplify the graphic to present the dependencies and part of the text as an independent paragraph for which the figure will be a graphic summary.
6. Section 7 should be named just "conclusions", since the discussion appeared earlier in the manuscript. The section should summarise the entire article, and I do not see the need to duplicate previously cited citations.
The authors have undertaken a very ambitious task of summarizing a significant area of ​​knowledge. I want to ask the authors whether section 3 should be included in this manuscript because the content of the work itself refers to the impact of microbiota on health and potential nutritional interventions, so I do not see the place in this review for additional weaving in a section referring to analytics. In my opinion, the section does not fit this manuscript.
Despite my comments, I believe this ambitious work should be accepted after making the necessary changes.
Author Response
Thank you for your valuable feedback. We are pleased to inform you that the manuscript has already been revised to address the points you raised. Below is an outline of the changes that have been implemented:
Minor suggestions:
- Abstract - I suggest the authors change the structure of the abstract into a coherent text without introducing additional sections.
We have carefully considered your suggestion and modified the abstract to meet your comment.
- line 67 – add citation
Done
- line 85 – citation after the end of the sentence
Done
- line 91 – add space before the citation
Done
- line 97 – authors should use full name type 2 diabetes Mellitus if the letter M appears in the abbreviation
Done
- line 190 - I suggest removing the quote here; it is still repeated in Table 3
- line 223 – as above
- line 634 – as above
Thank you for drawing our attention to this case, the quotes were deleted to avoid redundancy.
- table 5 - I suggest adding a separate column for citations
Done
- line 668 – change flora into the microbiome
Done
- line 680 - note as in the case of other tables regarding citations
- line 700 – move the citation to the end of the sentence
Done
- section 6.2 – lack of proper title for the section
Done
Major suggestions:
- line 124-136 – I would suggest that the authors enrich the description with specific examples, allowing for the identification of microorganisms, taking into account the type of substrate and its differentiating changes
Thank you for your valuable suggestion, we have expanded the description by writing the following examples: Page (3), Line (131)
“For example, Clostridium difficile can be isolated using specific anaerobic culture conditions with selective media containing fructose and cefoxitin, while Escherichia coli thrives in aerobic conditions with lactose-based media like MacConkey agar, allowing for differential identification based on substrate fermentation.”
- table 4 - the table should be a coherent summary of the content, allowing for efficient reading of the content; here in the table, we have collected text, which could be subsequent paragraphs. I suggest the authors change the table's structure and shorten the text or remove it and present the content in the form of text.
Thank you for the valuable comment, the table was reorganized and rewritten to meet the recommendations.
- Line 233 – 236 - the text seems to be written using AI, I suggest the authors to change the language to a less writerly and simpler one.
Thank you for this comment, the lines were rewritten. Page (8), Line (228-230)
“Recent mouse experiments show that bacterial species grow and use resources differently in the outer mucus layer versus the intestinal lumen, highlighting the need for careful sampling methods in microbiota analysis”
- Section 4.4 - I suggest moving the section to the beginning of section 4.3 as an introduction, it is more of an introduction or summary of section 4.3 than a standalone chapter 4.4
We fully acknowledge the reviewer’s comments regarding this section, it was used now as an introduction for section 4.3. Page (13), Line (375)
- Figure 2 - I appreciate the desire to present such a large amount of content in one graphic, but it is illegible, especially concerning metabolism. In addition, the name of the graphic itself is extended to the level of a separate paragraph and should be shortened. I suggest the authors simplify the graphic to present the dependencies and part of the text as an independent paragraph for which the figure will be a graphic summary.
Thank you for your valuable feedback on Figure 2. We simplified the graphic to enhance readability, especially in the metabolism section, and shortened its title for clarity. Additionally, we separated some of the detailed information into a standalone paragraph, allowing the figure to serve as a concise visual summary of the dependencies. Page (16), Line (527)
“The complex interaction between gut microbiota and human health involves various bacterial species, their metabolites, and their effects on host physiology. The gut microbiota comprises diverse bacterial species, including Akkermansia (Gram-negative, anaerobic), Roseburia (Gram-positive, anaerobic), non-pathogenic Escherichia coli, Bifidobacteria (Gram-positive, obligate anaerobes), and Lactobacilli (Gram-positive, facultative anaerobes). These bacteria produce several important metabolic compounds, including:
- Short-chain fatty acids (SCFAs) such as acetate, propionate, and butyrate.
- Amino acid metabolites like ammonia and indole.
- Essential vitamins, particularly vitamin K and B-group vitamins.
- Postbiotics, including extracellular polysaccharides.
- Fermentation metabolites such as lactate and glycerol.
- Various bioactive components, including bacteriocins
These bacterial metabolites significantly influence human health through multiple pathways, particularly in regulating metabolism (lipid, carbohydrate, and protein metabolism), maintaining intestinal barrier function, modulating inflammation, and regulating immune responses (Figure 2).”
- Section 7 should be named just "conclusions", since the discussion appeared earlier in the manuscript. The section should summarise the entire article, and I do not see the need to duplicate previously cited citations.
Thank you for your constructive comment. We have carefully considered your suggestion and kept this section as “conclusions” and avoided the repetition of the citation.
The authors have undertaken a very ambitious task of summarizing a significant area of ​​knowledge. I want to ask the authors whether section 3 should be included in this manuscript because the content of the work itself refers to the impact of microbiota on health and potential nutritional interventions, so I do not see the place in this review for additional weaving in a section referring to analytics. In my opinion, the section does not fit this manuscript.
Thank you for your valuable feedback regarding Section 3, "Measurement of Gut Microbiota." We appreciate your perspective on its relevance. While we have modified this section to ensure it does not detract from the main focus on microbiota's impact on health and nutritional interventions, we believe it remains essential. Understanding and analyzing microbiota composition is a crucial foundation for precision nutrition, supporting targeted dietary recommendations.
Despite my comments, I believe this ambitious work should be accepted after making the necessary changes.
Reviewer 2 Report
Comments and Suggestions for Authors
The authors summarized the progress of gut microbiota and precision nutrition and personalized medicine. The topic is cutting-edging and many investigators will be interested in the content. The manuscript exist the following questions:
1. What is precision nutrition? Please give a precise definition.
2. The part “3. Measurement of Gut Microbiota”. This section should not be the focus of this review and need to be simplified. It need not to list the advantages and disadvantages of different technical approaches, but simply describe them in the text.
3. Too many tables, too complex presentation. For example, Table 4, There are long paragraphs in the table, and it just needs to outline important key points.
4. “5.1. Regulation of Lipid Metabolism”,
“5.2. Regulation of Protein Metabolism”,
“5.3. Regulation of Carbohydrate Metabolism”
“5.4. Impact of Dietary Fibers”
“5.5. Impact of Prebiotics, Probiotics, Postbiotics, and Synbiotics”
“5.6. Impact of Chewing”
It's obvious that “5.1”, “5.2”, “5.3” and “5.4”, “5.5”, “5.6” is not s not a juxtaposition.
Part “5.4”, “5.5”, “5.6” should be integrated into “4.2. Dynamism and Influencing Factors”. They are influencing factors of gut microbiota.
5. For Figure 2, Some words in the picture are too small and can not read clearly; meanwhile, some word is too large that the size the text, for example, “bacteriocins”.
6. Minor errors, such as:
4.3. Impact of Gut Microbiota on other bodily sistems---4.3. Impact of Gut Microbiota on other body systems
Type 1 Diabetes (T1D)--- type 1 diabetes (T1D)
.[33,34]---- [33,34].
Comments on the Quality of English LanguageLanguage needs to be polished.
Author Response
Thank you for your valuable feedback. We are pleased to inform you that the manuscript has already been revised to address the points you raised. Below is an outline of the changes that have been implemented:
Reviewer 2:
The authors summarized the progress of gut microbiota and precision nutrition and personalized medicine. The topic is cutting-edging and many investigators will be interested in the content. The manuscript exist the following questions:
- What is precision nutrition? Please give a precise definition.
The definition of the Precision Nutrition was added on Page (2), Line (86).
“Precision nutrition is a tailored dietary strategy that considers an individual's unique genetic, environmental, and physiological factors to optimize health outcomes and prevent disease.”
- The part “3. Measurement of Gut Microbiota”. This section should not be the focus of this review and need to be simplified. It need not to list the advantages and disadvantages of different technical approaches, but simply describe them in the text.
Thank you for your valuable feedback regarding Section 3, the table was simplified as requested.
- Too many tables, too complex presentation. For example, Table 4, There are long paragraphs in the table, and it just needs to outline important key points.
Thank you for this comment, table 4 was reorganized and written in a direct informative way now and Table 5 was deleted.
- “5.1. Regulation of Lipid Metabolism”,
“5.2. Regulation of Protein Metabolism”,
“5.3. Regulation of Carbohydrate Metabolism”
“5.4. Impact of Dietary Fibers”
“5.5. Impact of Prebiotics, Probiotics, Postbiotics, and Synbiotics”
“5.6. Impact of Chewing”
It's obvious that “5.1”, “5.2”, “5.3” and “5.4”, “5.5”, “5.6” is not s not a juxtaposition.
Part “5.4”, “5.5”, “5.6” should be integrated into “4.2. Dynamism and Influencing Factors”. They are influencing factors of gut microbiota.
Thank you for the suggestion. We have integrated Parts “5.4,” “5.5,” and “5.6” into Section “4.2. Dynamism and Influencing Factors,” as they indeed represent factors that influence the gut microbiota. This restructuring provides a clearer and more cohesive discussion of the elements affecting microbiota composition.
- For Figure 2, Some words in the picture are too small and can not read clearly; meanwhile, some word is too large that the size the text, for example, “bacteriocins”.
Thank you for the insightful comment, the figure was modified to be more clear and homogenized.
- Minor errors, such as:
4.3. Impact of Gut Microbiota on other bodily sistems---4.3. Impact of Gut Microbiota on other body systems
Type 1 Diabetes (T1D)--- type 1 diabetes (T1D)
.[33,34]---- [33,34].
Thank you for your constructive comment. All of these corrections were done.
Comments on the Quality of English Language:
- Language needs to be polished.
The manuscript has been thoroughly revised for improving the language. We have ensured that the language is clear, precise, and free from inconsistencies, thereby improving the overall readability and scientific professionalism of the text.
Reviewer 3 Report
Comments and Suggestions for Authors
Thank you for submitting the manuscript "Unraveling the Gut Microbiota: Implications for Precision Nutrition and Personalized Medicine" to Nutrients. The manuscript is well written and the literature review seems to have been conducted in a comprehensive manner. I have a few minor observations to make.
- Consider including in item 4.1 how the phyla ratios behave, for example, Firmicutes/Bacteroidetes.
- I do not think that outlining the methods for identifying microorganisms in the microbiota is the objective of the work. This topic seems to me to be a bit outside the scope of the review.
- In addition, item 4 seems to me to be too general and could be included in other parts of the text.
Comments on the Quality of English LanguageThe English could be improved to more clearly express the research.
Author Response
Thank you for your valuable feedback. We are pleased to inform you that the manuscript has already been revised to address the points you raised. Below is an outline of the changes that have been implemented:
Reviewer 3:
Thank you for submitting the manuscript "Unraveling the Gut Microbiota: Implications for Precision Nutrition and Personalized Medicine" to Nutrients. The manuscript is well written and the literature review seems to have been conducted in a comprehensive manner. I have a few minor observations to make.
- Consider including in item 4.1 how the phyla ratios behave, for example, Firmicutes/Bacteroidetes.
Thank you for this valuable comment, the part concerning Firmicutes/Bacteroidetes was added on Page (8), Line (203)
“Recent research indicates that the ratios of these phyla, particularly the Firmicutes/Bacteroidetes (F/B) ratio, serve as important biomarkers of gut dysbiosis and are frequently cited in the literature as indicators of obesity [https://pubmed.ncbi.nlm.nih.gov/32438689/] [https://pubmed.ncbi.nlm.nih.gov/35889844/] Studies have shown that obese individuals typically exhibit a higher F/B ratio compared to their normal-weight counterparts, suggesting a link between this ratio and metabolic disorders . Conversely, lower F/B ratios have been associated with leaner phenotypes and may reflect a healthier gut microbiome [https://pubmed.ncbi.nlm.nih.gov/32438689/].
In clinical settings, the F/B ratio has been correlated with various health outcomes, including inflammation and metabolic syndrome [https://pubmed.ncbi.nlm.nih.gov/35889844/]. For instance, increased F/B ratios are observed in patients with cirrhosis, where they correlate with worse prognosis and higher mortality rates [https://doi.org/10.1016/j.clinsp.2024.100471]. These findings highlight the relevance of the F/B ratio not only as a marker of obesity but also as a potential indicator of overall gut health and disease susceptibility.”
- I do not think that outlining the methods for identifying microorganisms in the microbiota is the objective of the work. This topic seems to me to be a bit outside the scope of the review.
Thank you for your valuable feedback regarding Section 3, "Measurement of Gut Microbiota." We appreciate your perspective on its relevance. While we have modified this section to ensure it does not detract from the main focus on microbiota's impact on health and nutritional interventions, we believe it remains essential. Understanding and analyzing microbiota composition is a crucial foundation for precision nutrition, supporting targeted dietary recommendations.
- In addition, item 4 seems to me to be too general and could be included in other parts of the text.
Thank you for your insightful feedback. Table 4 was reorganized and written in a direct informative way
Comments on the Quality of English Language:
- The English could be improved to more clearly express the research.
The manuscript has been thoroughly revised for improving the language. We have ensured that the language is clear, precise, and free from inconsistencies, thereby improving the overall readability and scientific professionalism of the text.
Round 2
Reviewer 1 Report
Comments and Suggestions for Authors
I thank the authors for their efforts in providing comprehensive explanations and making the corrections I suggested. The manuscript is now suitable for publication in its current form.
Reviewer 2 Report
Comments and Suggestions for Authors
OK!